# De Quervain Tenosynovitis as a Risk Factor of New-Onset Adhesive Capsulitis: A Nationwide Cohort Study

**DOI:** 10.3390/healthcare11121758

**Published:** 2023-06-15

**Authors:** Pao Huang, Ching-I Hong, Chung-Chao Liang, Wen-Tien Wu, Jen-Hung Wang, Kuang-Ting Yeh

**Affiliations:** 1Department of Physical Medicine and Rehabilitation, Hualien Tzu Chi Hospital, Buddhist Tzu Chi Medical Foundation, Hualien 970, Taiwan; pao@tzuchi.com.tw (P.H.); stone@tzuchi.com.tw (C.-C.L.); 2Department of Special Education, National Dong Hwa University, Hualien 974, Taiwan; hor@gms.ndhu.edu.tw; 3School of Medicine, Tzu Chi University, Hualien 970, Taiwan; timwu@tzuchi.com.tw; 4Department of Orthopedics, Hualien Tzu Chi Hospital, Buddhist Tzu Chi Medical Foundation, Hualien 970, Taiwan; 5Institute of Medical Sciences, Tzu Chi University, Hualien 970, Taiwan; 6Department of Medical Research, Hualien Tzu Chi Hospital, Buddhist Tzu Chi Medical Foundation, Hualien 970, Taiwan; paulwang@tzuchi.com.tw; 7Graduate Institute of Clinical Pharmacy, Tzu Chi University, Hualien 970, Taiwan

**Keywords:** de Quervain tenosynovitis, adhesive capsulitis of shoulder, occupational therapy, rehabilitation, nationwide cohort database

## Abstract

This study investigated the association of de Quervain tenosynovitis (DQT) with subsequent adhesive capsulitis (AC) development. Patients with DQT between 2001 and 2017 from the Taiwan National Health Insurance Research Database were the DQT cohort. The 1:1 propensity score matching method was applied for creating control cohort. The primary outcome was defined as new-onset of AC at least 1 year after the date of confirmed diagnosis of DQT. In total, 32,048 patients with mean age 45.3 years were included. DQT was significantly positively associated with risk of new-onset AC after adjustment for baseline characteristics. Furthermore, severe DQT requiring rehabilitation was positively associated with risk of new-onset AC. In addition, male gender and age under 40 may be additional risk factors for new-onset AC, compared to female gender and age over 40. Cumulative incidence of AC after 17 years was 24.1% among patients with severe DQT requiring rehabilitation and was 20.8% among patients with DQT without rehabilitation. This is the first population-based study to demonstrate an association between DQT and new-onset AC. The findings recommend that preventive occupational therapy, including active modification for the shoulder joint and adjustments to daily activities, may be necessary for patients with DQT to reduce their risk of developing AC.

## 1. Introduction

De Quervain tenosynovitis (DQT) is a common wrist pathology occurring in the first dorsal compartment and is characterized by pain that is caused by the resisted gliding of the abductor pollicis longus and the extensor pollicis brevis tendons in the fibro-osseous canal [1]. A study involving 11,332 patients with DQT demonstrated that women have a significantly higher rate of DQT, at 2.8 cases per 1000 person-years, compared with men at 0.6 per 1000 person-years [2]. DQT can be managed nonsurgically through a combination of rest, ice, splinting, and physical therapy [3]. Nonsurgical treatments should be tried first, and surgery should only be considered if symptoms persist or worsen [3]. Fedorczyk et al. note that histopathology studies have revealed the presence of abnormal tissue in tendons affected by tenosynovitis and that this tissue may respond to conservative treatments such as eccentric exercise [4]. Robinson et al. describe a successful rehabilitation program for a cellist with DQT and intersection syndrome, which included both nonsurgical interventions and surgery [5]. Papa et al. present a case report of a patient with DQT who responded well to conservative management with chiropractic care [6]. Jaworski et al. outline rehabilitation strategies for athletes with wrist and hand injuries, including exercises to improve range of motion and strength [7]. Shoulder adhesive capsulitis, commonly known as frozen shoulder, is a condition that causes pain and restricted movement in the shoulder joint. Diagnosis is typically made based on physical examination and imaging studies. Typical physical examination findings include deep and aching pain, worse pain at night, presentation of pain arc, stiffness and restricted range of motion, and wasting or atrophy of shoulder muscles. Typical imaging study findings include joint space narrowing on shoulder X-rays and capsular thickening change on magnetic resonance imaging or ultrasound studies [8]. According to Ramirez et al., initial treatment for shoulder adhesive capsulitis should focus on reducing pain and improving range of motion through a combination of physical therapy and nonsteroidal anti-inflammatory drugs [8]. Redler and Dennis et al. note that early corticosteroid injection may shorten the duration of symptoms in adhesive capsulitis; however, patients should be aware that anti-inflammatories and corticosteroid injections do not provide a cure but rather alleviate discomfort during necessary physical therapy, while extracorporeal shock wave therapy may be beneficial for diabetic patients experiencing metabolic issues from corticosteroid injection [9]. D’Orsi et al. review the available treatment options for shoulder adhesive capsulitis, including both conservative and surgical approaches, and suggest that a multimodal treatment plan that includes physical therapy, nonsteroidal anti-inflammatory drugs, and steroid injections may be most effective [10]. Harris et al. conducted a systematic review of studies on the use of intra-articular hyaluronate for the treatment of shoulder adhesive capsulitis and found that, while the evidence is limited, there is some support for this approach in certain patient populations [11]. Shoulder adhesive capsulitis can also occur following shoulder surgery or trauma [12]. Previous studies have reported the epidemiology and risk factors for common disorders of the upper extremities, including DQT and shoulder adhesive capsulitis, but have seldom reported the correlation between DQT and shoulder adhesive capsulitis [4,13,14]. We previously reported the correlation between trigger finger and carpal tunnel syndrome and found the subsequent incidence of one disorder after treatment of the other [15,16]. In this study, we investigated the correlation between DQT and subsequent shoulder adhesive capsulitis using a nationwide cohort database.

## 2. Materials and Methods

This study was approved by the Research Ethics Committee of Hualien Tzu Chi Hospital (IRB 108-242-C). Data were obtained from the Taiwan National Health Insurance Research Database, which is maintained by the National Health Insurance Administration, Ministry of Health and Welfare, and managed by the National Health Research Institutes. In total, 18,249 patients with DQT between 1 January 2001, and 31 December 2017, were included as the DQT group. In addition, 1,981,875 patients without DQT during the same period were enrolled from the general population of the Taiwan National Health Insurance Research Database through random sampling as the control group (Figure 1). Patients who were younger than 20 years or who had a prior diagnosis of shoulder adhesive capsulitis before their index date (the date of diagnosis of DQT) were excluded. We used 1:1 propensity score matching for further analysis. The outcome of this study was the subsequent development of shoulder adhesive capsulitis 1 year or more after the date of DQT diagnosis. We then divided the DQT group into two subgroups based on the requirement of rehabilitation, comparing the severity of DQT with the control group.

All statistical analyses were performed using SAS version 9.4 and Stata version 16 (SAS Institute, Cary, NC, USA). Continuous variables are summarized as mean (standard deviation), and categorical variables are listed as number and percentage. Between-group comparisons of continuous variables were performed using Student’s *t* test, and categorical variables were assessed using either a chi-square test or Fisher’s exact test. Data were evaluated using a log-rank test and univariate and multivariate Cox regression analyses. Survival curves were estimated according to the Kaplan–Meier procedure and groups were compared with the log-rank test. Statistical significance was indicated by *p* values of <0.05.

## 3. Results

In total, 16,026 patients with DQT were included. The mean age of these patients was 45.3 (13.6) years. In total, 3886 (24.3%) patients were men and 12,138 (75.7%) patients were women. In total, 10,180 (63.5%) patients were 40 years or older, and 8108 (50.6%) patients required DQT-related rehabilitation (Table 1). In addition, 16,024 patients without DQT were included. The two groups were adequately balanced in terms of baseline characteristics of age, gender, and percentage of comorbidities (Table 1). The incidence of shoulder adhesive capsulitis among patients with DQT was 15.8 per 1000 person-years. After adjustment for all the baseline characteristics, the risk of new-onset shoulder adhesive capsulitis was significantly higher among patients with DQT than among those without DQT (adjusted hazard ratio [aHR] = 1.68, 95% confidence interval [CI] = 1.54–1.82, *p* < 0.001; Table 2). Specifically, patients requiring rehabilitation had a higher risk of shoulder adhesive capsulitis (adjusted hazard ratio [aHR] = 2.12, 95% confidence interval [CI] = 1.93–2.33, *p* < 0.001), while those not requiring rehabilitation had a lower but still significant risk (aHR = 1.30, 95% CI = 1.18–1.44, *p* < 0.001; Table 3). Moreover, among patients with DQT, those requiring rehabilitation had a higher risk of shoulder adhesive capsulitis compared to those not requiring rehabilitation (aHR = 1.62, 95% CI = 1.46–1.80, *p* < 0.001; Table 3). In a subgroup analysis by age, the presence of DQT was positively associated with the risk of new-onset shoulder adhesive capsulitis among patients younger than 40 years (aHR = 1.87, 95% CI = 1.47–2.38, *p* < 0.001) and among those 40 years or older (aHR = 1.67, 95% CI = 1.53–1.83, *p* < 0.001; Table 4). Similarly, in a subgroup analysis by gender, the presence of DQT was positively associated with the risk of new-onset shoulder adhesive capsulitis among men (aHR = 2.10, 95% CI = 1.72–2.56, *p* < 0.001) and among women (aHR = 1.62, 95% CI = 1.48–1.77, *p* < 0.001; Table 4). The proportion of patients with DQT with new-onset shoulder adhesive capsulitis increased linearly over time. The cumulative incidence of shoulder adhesive capsulitis after 17 years was 24.1% among patients with DQT who required rehabilitation and was 20.8% among patients with DQT who had not required rehabilitation (Figure 2).

## 4. Discussion

We found that DQT was positively associated with the cumulative risk of new-onset shoulder adhesive capsulitis. DQT is a common pathology of the radial side of the wrist caused by stenosing tenosynovitis of the extensor pollicis brevis and abductor pollicis longus tendons [1]. The prevalence of DQT is higher in women than in men and it typically affects middle-aged individuals [3]. Petit Le Manac’h et al. found that the prevalence of DQT among the working population was 1.2%, and its prevalence among women and men was 2.1% and 0.6%, respectively [17]. The prevalence of shoulder adhesive capsulitis is between 2% and 5%, and shoulder adhesive capsulitis is more common in women than in men in the working population. Studies have reported several risk factors for shoulder adhesive capsulitis, including cervical disc discectomy, upper extremity trauma, thyroid disease, diabetes, Dupuytren’s contracture, breast cancer, and autoimmune disease [18]. The main causes of shoulder adhesive capsulitis are inflammatory contracture of the shoulder capsule caused by the accumulation of inflammatory cytokines and the production of fibrous-inducing factors, such as TGF-β1 and platelet-derived growth factor [19]. DQT was also notably caused by the accumulation of mucopolysaccharide as a signal of degeneration based on histopathology findings, but the inflammation may aggravate the symptom of the disease [20]. The effects of these inflammatory cytokines may be the connection between the two diseases. In addition, shoulder adhesive capsulitis may result from overprotection and limited shoulder use after surgery [21]. From the study results we also found that severe DQT that required rehabilitation was positively associated with the risk of new-onset shoulder adhesive capsulitis among patients with DQT. Since the causes of correlation between DQT and shoulder adhesive capsulitis are not fully understood, we proposed two possible mechanisms relating to these correlations between DQT and shoulder adhesive capsulitis.

First, a potential cause of shoulder adhesive capsulitis is the underutilization of the shoulder joint due to limb pain caused by ipsilateral DQT. Patients with DQT may modify their posture or behavior in response to hand movement difficulties or pain, potentially exacerbating the progression of shoulder stiffness [22]. The severity of DQT may be positively correlated with the risk of new-onset shoulder adhesive capsulitis because these two diseases share several risk factors. Patients with a history of one condition may be more prone to developing other musculoskeletal conditions, which could increase the risk of shoulder adhesive capsulitis. These may explain that patients with more severe DQT represented by those who required further rehabilitation may have a higher rate predisposing shoulder adhesive capsulitis. When a distal joint such as the thumb joint is injured and painful, the proximal joint, such as the shoulder joint, is often used to assist with daily activities, which may cause further injury of the shoulder, such as sprain injury or impingement disease, then further aggravating the development of shoulder adhesive capsulitis [23].

Secondly, a rehabilitation program and conservative treatment of DQT without considering shoulder condition may also play a role in the development of shoulder stiffness. Inadequate rehabilitation protocol was claimed to cause progressive shoulder stiffness after arthroscopic rotator cuff tendon repair [24]. Denard et al. observed that the 6-week immobilization protocol may reduce the incidence of postoperative shoulder stiffness in patients who received shoulder surgery [25]. Rehabilitation of patients with DQT involves protection with a splint or brace and the range of motion protocol of the wrist and thumb. The protection of the upper limb and the rehabilitation program without considering the influence of shoulder movement may induce shoulder adhesive capsulitis based on the above information. Patients who receive a rehabilitation program without considering shoulder movement may further accidentally sprain the tendons or ligaments of their shoulder joint, causing pain, which may lead to limited use of their shoulder joint, causing the development of shoulder adhesive capsulitis and shoulder stiffness. Therefore, specialized shoulder physiotherapy may be recommended for the patients to prevent them from the development of postoperative shoulder stiffness [26]. This could involve incorporating exercises and activities that promote proper posture and mechanics. Additionally, proper supervision and monitoring during rehabilitation may help prevent patients from further injury or overly limited use of their shoulder joints [27,28]. We believe that specialized shoulder physiotherapy should be offered to patients with severe DQT who need rehabilitation to prevent the development of additional shoulder problems. Incorporating exercises that specifically target the shoulder joint, including rotator cuff strengthening exercises, shoulder range-of-motion exercises, and shoulder stabilization exercises may help improve shoulder strength and flexibility, reducing the risk of developing shoulder adhesive capsulitis [29]. Occupational therapy interventions, including giving instructions on joint protection and energy conservation, use of assistive devices, and the provision of splints, may be helpful for treatment of DQT and the prevention of further shoulder adhesive capsulitis development [30]. Patients may benefit from education on proper ergonomics and body mechanics during daily activities, particularly those that involve the use of the wrist and hand [31]. This may help reduce the incidental rate of shoulder injury, minimize the risk of inflammation, and decrease the risk of developing subsequent shoulder adhesive capsulitis.

Since severe DQT requiring rehabilitation has a relatively high incidence of subsequent shoulder adhesive capsulitis, we suggest the physicians and the physical therapists should include shoulder movement into the rehabilitation program for DQT. During the recovery phase (2–6 weeks after acute attack of DQT) it should focus on restoring function to the affected wrist and hand, and may initiate prevention measures for shoulder stiffness, including a gentle range of motion exercises for the shoulder (flexion, extension, abduction, and internal and external rotation, etc.) and maintain shoulder mobility to prevent stiffness and ensure proper posture during daily activities [32]. After 6 weeks, in addition to the preventing procedures of recurrence of DQT, strengthening exercises for the shoulder are advised, particularly for the rotator cuff and scapular muscles, to keep the shoulder mobility and prevent it from accidental injury [33]. We also suggest that the patient should also maintain proper posture during all activities, keeping the spine aligned and the shoulders relaxed so that they are easily sprained or injured.

Occupational therapy plays a crucial role in the treatment of DQT, a painful condition that affects the tendons on the thumb side of the wrist [30]. When the abductor pollicis longus and the extensor pollicis brevis around the base of the thumb become swollen or inflamed, the pain and discomfort in the wrist and lower thumb may extend to the forearm and shoulder if left untreated [17]. Occupational therapists provide individualized interventions aimed at reducing pain and inflammation, and improving functional abilities. Firstly, it is important to mention that the main focus of occupational therapy in treating DQT is to promote healing, reduce pain, and restore function. This is usually achieved by teaching patients ergonomics and body mechanics, and joint protection techniques, and promoting self-care activities [31]. One of the first steps in treatment often includes rest and immobilization. The occupational therapist may recommend wearing a splint or brace to restrict movement and reduce inflammation. It is important to avoid activities that exacerbate symptoms, such as gripping, pinching, or repetitive hand and wrist movements. In terms of ergonomics, occupational therapists provide advice on modifying workstations to alleviate strain on the wrist and thumb [34]. This may involve recommending the use of ergonomic equipment, such as wrist rests, or suggesting adjustments in how tasks are performed. For example, the therapist might suggest different ways of holding or manipulating objects to reduce stress on the affected tendons. Education about joint protection techniques is another important aspect of treatment [35]. The therapist may need to teach strategies to reduce stress on the thumb and wrist, such as using larger joints for tasks and avoiding positions that could exacerbate symptoms. These techniques not only help manage current symptoms but also prevent future flare-ups. Exercises designed to improve strength and flexibility also form part of occupational therapy [30]. Gentle range-of-motion exercises, strengthening exercises, and stretches can help improve function, reduce pain, and prevent further injury. These are typically performed under the guidance of the therapist and then integrated into a home exercise program. Furthermore, occupational therapists often teach pain management techniques [35]. These could include heat or cold therapy, massage, or relaxation techniques to help control pain. Therapists might also suggest modifications in self-care activities, such as bathing, dressing, cooking, and other tasks, to help manage pain and prevent aggravation of symptoms.

Because pain or discomfort from DQT can extend up the forearm and potentially increase the risk of shoulder injury, causing the incidence of shoulder adhesive capsulitis, occupational therapists can help address these secondary issues by teaching patients correct postural habits to reduce strain on the shoulder [36]. This includes maintaining a neutral spine, avoiding hunched shoulders, and keeping the elbow close to the body during activities. Therapists may also provide exercises to strengthen the shoulder and improve posture, and suggest modifications in daily activities to reduce risk of injury to the shoulder [37]. By focusing on individualized treatment plans, occupational therapists can help patients reduce pain, improve function, and maintain their quality of life. Their multidimensional approach not only helps in the immediate recovery from this painful condition, but it also provides patients with the knowledge and skills to manage their symptoms and prevent recurrence of DQT, or incidence of other tendinitis and tenosynovitis of the upper limb in the future.

In this study, female gender and age greater than 40 years were also found to be correlated with the occurrence of new-onset shoulder adhesive capsulitis. Wolf et al. conducted a large population study in 2009 and found that the risk factors for DQT in the young and active population included Black race, female gender, and age more than 40 years [2]. Benites-Zapata et al. in 2021 also found a higher prevalence of DQT symptomatology in people aged between 18 and 25 years with problematic smartphone use [38] and most of their study participants were women. Kingston et al. revealed that younger patients and racial minorities were significantly more likely to be diagnosed with shoulder adhesive capsulitis [39]. Our study results are not different from these results because, although male gender and age less than 40 years are not common risk factors in the literature, if a patient is diagnosed with DQT, their treatment may be more difficult, which is more likely to result in subsequent shoulder adhesive capsulitis. Few studies have discussed the association of DQT with the development of shoulder adhesive capsulitis. Mandiroglu et al. in 2021 revealed that DQT is more prevalent among patients with idiopathic carpal tunnel syndrome than in the general population, and that the early identification and diagnosis of both pathologies may have a positive influence on treatment [40]. For men younger than 40 years undergoing treatment for DQT, monitoring for the development of shoulder adhesive capsulitis is recommended. Preventive occupational therapy with active modification for shoulder joints and adjustments for daily activities are recommended for the two high-risk subgroups (male gender and age less than 40 years) of patients with DQT to reduce the risk of shoulder adhesive capsulitis and improve their quality of life.

This study provides important insights into the potential association between DQT and subsequent shoulder adhesive capsulitis. One of the key advantages of this study is the large sample size used to analyze the risk of new-onset shoulder adhesive capsulitis among patients with DQT. This is the first study to report on the correlation between these two conditions, providing valuable information for clinicians and researchers. Another advantage of this study is the use of the comprehensive coverage of the National Health Insurance system, which includes more than 96% of the population. This reduces the likelihood of selection and nonresponse biases, providing a more accurate representation of the population. However, there are several limitations to this study that should be considered. One limitation is that the severity of DQT and shoulder adhesive capsulitis could not be defined, as the database does not include information on symptoms and physical findings. This limits the ability to fully understand the relationship between these conditions and the potential mechanisms underlying the association. Another limitation is that patients may have undergone other types of conservative treatment unsupported by health insurance, such as acupuncture or massage therapy, which could impact the outcomes. Additionally, detailed employment information was not available in the database, which may impact the development of these conditions. Furthermore, the inability to determine whether the frozen shoulder affects the same side (ipsilateral) or opposite side (contralateral) as the DQT can greatly impede the establishment of causal hypotheses based on available data. However, obtaining comprehensive information on the progression of both conditions through detailed clinical case studies may better elucidate their interrelationships as functional components of upper limb movement. Despite these limitations, this study provides important insights into the possible associations between DQT and subsequent shoulder adhesive capsulitis. The presence of DQT provides an opportunity for clinicians to inform patients of the symptoms of adhesive capsulitis and recommend regular shoulder examinations for the purpose of early detection of adhesive capsulitis to maximize implemented treatment outcomes. The identification of this association between these two pathologies also helps develop preventive programs for shoulder joint health during the rehabilitation for patients with a more severe degree of DQT. Since there is no evidence supporting DQT as a cause of or contribution to the cause of shoulder adhesive capsulitis at this point in time, future studies also should be conducted to investigate the effects of alterations to rehabilitation programs and occupational therapy on the risk of shoulder adhesive capsulitis among patients with DQT. This will allow us to understand how exactly shoulder and thumb cooperate with each other dynamically in different situations.

## 5. Conclusions

In this nationwide cohort study, researchers found a positive association between DQT and the risk of new-onset shoulder adhesive capsulitis. Severe DQT that required rehabilitation was also positively associated with the risk of new-onset shoulder adhesive capsulitis. Among patients with DQT, age less than 40 years and male gender may also be risk factors for new-onset shoulder adhesive capsulitis. Overall, this study highlights the importance of identifying and addressing risk factors for shoulder adhesive capsulitis among patients with DQT. By implementing targeted preventive measures, including rehabilitation protocol, patient education incorporating shoulder strengthening, physiotherapy, and related occupational therapy, clinicians may be able to reduce the incidence and severity of shoulder adhesive capsulitis in this patient population.

## Figures and Tables

**Figure 1 healthcare-11-01758-f001:**
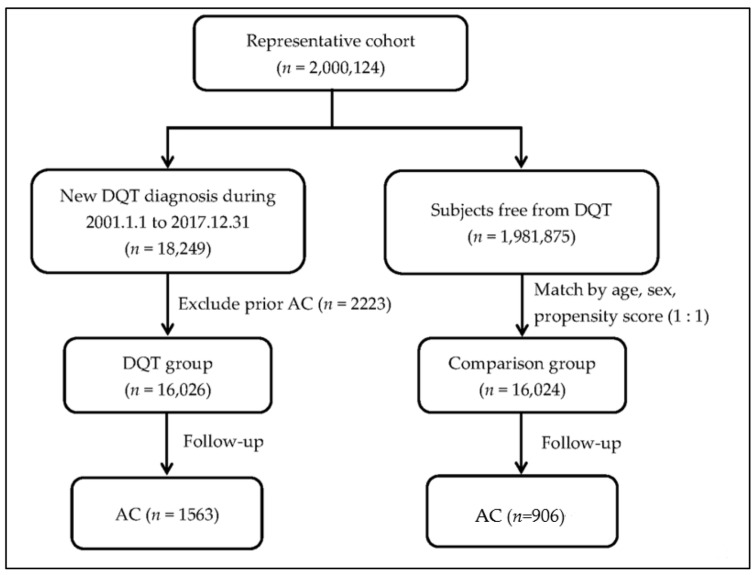
The study flow chart of the inclusion of both cohort groups. DQT: de Quervain tenosynovitis; AC: shoulder adhesive capsulitis.

**Figure 2 healthcare-11-01758-f002:**
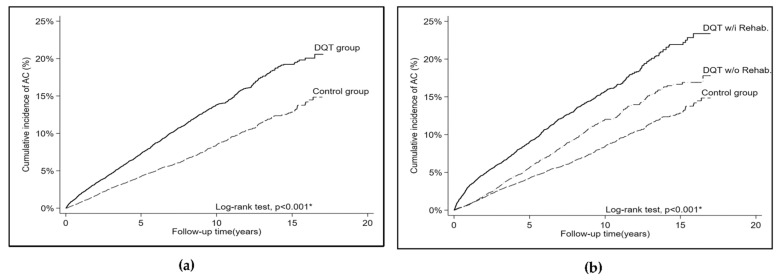
(**a**) The proportion of subsequent adhesive capsulitis of the de Quervain tenosynovitis group and the control group. (**b**) The proportion of subsequent adhesive capsulitis of the de Quervain tenosynovitis with rehabilitation group, the de Quervain tenosynovitis without rehabilitation group, and the control group. * A value of *p* < 0.05 was considered statistically significant after test.

**Table 1 healthcare-11-01758-t001:** Baseline characteristics and comorbidity of the cohorts (*n* = 32,048).

Variables	Control (*n* = 16,024)	De Quervain Tenosynovitis (*n* = 16,024)	*p*-Value
Age (years)	45.3 ± 13.6	45.3 ± 13.7	0.960
Age group			0.966
<20 years	277 (1.7%)	280 (1.8%)	
20–40 years	5567 (34.7%)	5583 (34.8%)	
40–60 years	7882 (49.2%)	7840 (48.9%)	
≥60 years	2298 (14.3%)	2321 (14.5%)	
Age group			0.826
<40 years	5844 (36.5%)	5863 (36.6%)	
≥40 years	10,180 (63.5%)	10,161 (63.4%)	
Gender			0.815
Male	3886 (24.3%)	3904 (24.4%)	
Female	12,138 (75.7%)	12,120 (75.6%)	
Rehabilitation (%)	NA	8108 (50.6%)	
Comorbidity			
Hypertension	2438 (15.2%)	2416 (15.1%)	0.732
Diabetes mellitus	1274 (8.0%)	1261 (7.9%)	0.788
Hyperlipidemia	1721 (10.7%)	1702 (10.6%)	0.731
Coronary artery disease	503 (3.1%)	527 (3.3%)	0.447
Chronic liver disease	684 (4.3%)	695 (4.3%)	0.762
Hyperthyroidism	87 (0.5%)	104 (0.7%)	0.217

Data are presented as *n* and percentage.

**Table 2 healthcare-11-01758-t002:** Risk of adhesive capsulitis in patients with and without de Quervain tenosynovitis (*n* = 32,048).

Variables	De Quervain Tenosynovitis	
	Yes	No
Patient numbers	16,024	16,024
Adhesive capsulitis of shoulder cases	1563	906
Person-years	98,842	101,683
Incidence rate ^a^	15.8	8.9
Univariate model		
Crude HR (95% CI)	1.68 (1.55–1.83)	1 (ref.)
*p*-value	<0.001 *	
Multivariate model ^b^		
aHR (95% CI)	1.68 (1.54–1.82)	1 (ref.)
*p*-value	<0.001 *	

* *p*-value < 0.05 was considered statistically significant after test. HR: hazard ratio; aHR: adjusted hazard ratio; CI: confidence interval; ref: reference. ^a^ Per 1000 person-years. ^b^ Multivariate Cox proportional hazard regression model with adjustment for all baseline characteristics shown in Table 1.

**Table 3 healthcare-11-01758-t003:** Risk of shoulder adhesive capsulitis in patients with de Quervain tenosynovitis who did or did not require rehabilitation (*n* = 32,048).

Variables	De Quervain Tenosynovitis		Control
	W/I Rehabilitation	W/O Rehabilitation	
Patient numbers	7860	8164	16,024
Adhesive capsulitis of shoulder cases	910	653	906
Person-years	46,819	52,023	101,683
Incidence rate ^a^	19.4	12.6	8.9
Univariate model			
Crude HR (95% CI)	2.05 (1.87–2.25)	1.35 (1.22–1.50)	1 (ref.)
*p*-value	<0.001 *	<0.001 *	
Model 1 ^b^			
aHR (95% CI)	2.12 (1.93–2.33)	1.30 (1.18–1.44)	1 (ref.)
*p*-value	<0.001 *	<0.001 *	
Model 2 ^c^			
aHR (95% CI)	1.62 (1.46–1.80)	1 (ref.)	
*p*-value	<0.001 *		

* *p*-value < 0.05 was considered statistically significant after test. aHR: adjusted hazard ratio; CI: confidence interval; ref: reference. A total of 16,024 patients with de Quervain tenosynovitis were classified according to requiring rehabilitation or not. ^a^ Per 1000 person-years. ^b^ Model 1: aHR was calculated by using the control cohort as the reference group in a multivariate Cox proportional hazard regression model adjusting for all baseline characteristics in Table 2. ^c^ Model 2: aHR was calculated by using de Quervain tenosynovitis patients who did not require rehabilitation as the reference group in a multivariate Cox proportional hazard regression model adjusting for all baseline characteristics in Table 1.

**Table 4 healthcare-11-01758-t004:** Subgroup analysis of Cox’s regression model for the association between de Quervain tenosynovitis and adhesive capsulitis of shoulder.

Variables	Crude HR ^a^ (95% CI)	*p*-Value	Adjusted HR ^a^ (95% CI)	*p*-Value	*p* for Interaction
Main model					
Control	1.00		1.00		
De Quervain tenosynovitis	1.68 (1.55–1.83)	<0.001 *	1.68 (1.54–1.82)	<0.001 *	
Age					
<40 years					
Control	1.00		1.00		
De Quervain tenosynovitis	1.88 (1.47–2.39)	<0.001 *	1.87 (1.47–2.38)	<0.001 *	0.415
≥40 years					
Control	1.00		1.00		
De Quervain tenosynovitis	1.67 (1.53–1.82)	<0.001 *	1.67 (1.53–1.83)	<0.001 *	
Gender					
Male					
Control	1.00		1.00		
De Quervain tenosynovitis	2.09 (1.72–2.55)	<0.001 *	2.10 (1.72–2.56)	<0.001 *	0.021 *
Female					
Control	1.00		1.00		
De Quervain tenosynovitis	1.60 (1.46–1.76)	<0.001 *	1.62 (1.48–1.77)	<0.001 *	

* *p*-value < 0.05 was considered statistically significant after test. HR: hazard ratio; CI: confidence interval. ^a^ Cox’s proportional hazards model.

## Data Availability

Data is contained within the article.

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
