# Peer review of "De Quervain Tenosynovitis as a Risk Factor of New-Onset Adhesive Capsulitis: A Nationwide Cohort Study"

_healthcare, 2023, doi:10.3390/healthcare11121758_

Round 1
Reviewer 1 Report
Authors, This study is an incredible undertaking and the beginning of many studies that can follow to further understand the correlations identified. Amazing job and well-written manuscript.
Self-citation:
Huang P - 0/32
Hong CI - 0/32
Liang CC - 0/32
Wu WT - 2/32
Wang JH - 2/32
Yeh KT - 2/32
COMMENT: In the Abstract, de Quervain tenosynovitis is abbreviated as DQT. However, in the rest of the manuscript, it is spelled out. de Quervain tenosynovitis should be abbreviated at the first usage in the Introduction and then used as DQT throughout the manuscript.
Introduction, lines 41-43
de Quervain’s tenosynovitis can be managed nonsurgically through a combination of rest, ice, splinting, and physical therapy. [Source?]
Introduction, line 55
Diagnosis is typically made based on physical examination and imaging studies.
COMMENT: What are the gold standard findings and from what examination procedures? (name and findings of orthopedic tests and radiographs)
Introduction, lines 58-60
Redler and Dennis et al. note that steroid injections may also be used to manage symptoms, but caution that these should be used judiciously and only in combination with physical therapy [9].
COMMENT: Why should caution be used? What were the downsides of steroid use noted in the study?
Results, lines 122-123
COMMENT: You wrote "de Quervain’s tenosynovitis" (several times in the Results section) as opposed to other times in the manuscript where you referred to it as de Quervain tenosynovitis. I recommend you stick with one name throughout the manuscript if you insist on writing it out each time (subtle difference, but for the sake of consistency). See tables as well. This should be DQT anyways (see comment above about abbreviation).
Discussion, line 187
shoulder adhesive capsulitis may...
COMMENT: Shoulder should have a capital S.
Discussion, lines 187-191
shoulder adhesive capsulitis may occur on the ipsilateral side of the upper limb in patients with de Quervain’s tenosynovitis due to pain-related limited use of the limb. To compensate for difficulty or pain with hand movement, patients with de Quervain’s tenosynovitis may alter their posture or behavior, further complicating the development of shoulder soreness or stiffness.
COMMENT: Is there any source to support the notion that DQT leads to a lack of use of the shoulder? It is worth noting that you contradict this argument many times indicating that it can lead to overuse of the shoulder.
Discussion, lines 193-194
Both de Quervain's tenosynovitis and shoulder adhesive capsulitis are caused by inflammation of the affected tissues.
COMMENT: This is not true. Inflammation is a symptom of the conditions. Adhesive capsulitis is most likely caused by lack of use of the shoulder (immobility and reduced mobility) and DQT is most likely caused by overuse of the thumb and(or) wrist (repetitive movements, video games).
Discussion, lines 196-199
Patients with de Quervain's tenosynovitis may compensate for their limited wrist and thumb movement by overusing their shoulder joint, which could lead to inflammation and subsequent shoulder adhesive capsulitis.
and lines 218-220
This rehabilitation may burden the shoulder during movement and increase immobility of the affected upper limb during rest, which may induce shoulder adhesive capsulitis.
and lines 226-230
Patients with de Quervain's tenosynovitis may compensate for their limited wrist and thumb movement by overusing their shoulder joint, which could lead to inflammation and subsequent shoulder adhesive capsulitis. This compensatory movement pattern may persist even after rehabilitation, as patients may continue to use their shoulder joint more than they would otherwise.
and lines 236-239
Patients undergoing rehabilitation for de Quervain's tenosynovitis may be performing exercises and activities that place additional stress on their shoulder joint, which could lead to inflammation and subsequent shoulder adhesive capsulitis.
COMMENT: Adhesive capsulitis is not known for being an overuse injury, but on the contrary, a lack of use injury. This explanation does not make sense as is. Please provide a reputable source.
Discussion, lines 194-195
...it is possible that the inflammatory response may spread from one area of the body to another. [Source?]
and lines 233-235
This increased inflammation and damage to the affected tissues may increase the risk of developing subsequent shoulder adhesive capsulitis. [Source?]
Discussion, lines 252-256
When a distal joint such as the thumb joint is injured and painful, the proximal joint, such as the shoulder joint, is often used to assist with daily activities, which may cause further impingement syndrome or inflammation, further aggravating the development of shoulder adhesive capsulitis. [Source?]
COMMENT: If no source can be found, perhaps you can find any evidence supporting this notion for other comparable areas of the body (ie the big toe and the hip, perhaps in gouty arthritic patients?)
Discussion, lines 281-284
While the exact mechanisms underlying the association between de Quervain's tenosynovitis and subsequent shoulder adhesive capsulitis are not fully understood, there are several preventive rehabilitation procedures that may help reduce the risk of developing shoulder adhesive capsulitis in patients with de Quervain's tenosynovitis.
COMMENT: 1) The exact mechanisms underlying the association between DQT and subsequent shoulder adhesive capsulitis are not understood at all. This appears to be conjecture that is not supported by any evidence and seems contradictory to the evidence of the currently understood mechanism of injury for adhesive capsulitis. 2) There are no preventive rehabilitation procedures that may help reduce the risk of developing shoulder adhesive capsulitis in patients with de Quervain's tenosynovitis. This is the first study investigating this subject. Nothing has been investigated otherwise. Again, it is conjecture. Any recommendations need to be accompanied with a source of previous studies and should be more of a recommendation for future studies involving DQT patients as opposed to recommendations for treatment without evidence to support them. The following exemplifies this comment (Discussion, lines 331-337):
...this study provides important insights into the possible associations between de Quervain's tenosynovitis and subsequent shoulder adhesive capsulitis. The identification of this association provides an opportunity for clinicians to develop preventive programs for shoulder joint health during the rehabilitation of more severe cases of de Quervain's tenosynovitis. Further studies should be conducted to investigate the effects of alterations to rehabilitation programs and occupational therapy on the risk of shoulder adhesive capsulitis among patients with de Quervain's tenosynovitis.
COMMENT: It should also be noted that the presence of DQT provides an opportunity for clinicians to inform patients of the symptoms of adhesive capsulitis and recommend regular shoulder examinations for the purpose of early detection of adhesive capsulitis to maximize implemented treatment outcomes.
IMPORTANT: Your proposed mechanism of adhesive capsulitis injury in patients with DQT is extremely important and cannot be unfounded. Sources are needed for your comments throughout the Discussion (see above). And, if there is nothing to support your thoughts, that should be noted. You may need to write something along the lines of:
There is no evidence supporting DQT as a cause of or contribution to the cause of adhesive capsulitis at this point in time. More research is necessary to investigate the relationship between DQT and adhesive capsulitis. To better understand this relationship, future studies need to focus on... This will allow...
Amazing job and well-written manuscript. Very few edits regarding the quality of English Language.
Author Response
Authors, This study is an incredible undertaking and the beginning of many studies that can follow to further understand the correlations identified. Amazing job and well-written manuscript.
Self-citation:
Huang P - 0/32
Hong CI - 0/32
Liang CC - 0/32
Wu WT - 2/32
Wang JH - 2/32
Yeh KT - 2/32
Ans: Thank you very much for your encouragement.
COMMENT: In the Abstract, de Quervain tenosynovitis is abbreviated as DQT. However, in the rest of the manuscript, it is spelled out. de Quervain tenosynovitis should be abbreviated at the first usage in the Introduction and then used as DQT throughout the manuscript.
Ans: Thank you for your reminder. Because the submission guideline of Healthcare should be at least 4000 words for original article submission. We would like to keep de Quervain tenosynovitis for reaching the minimal word amounts requirement. I have also modified our manuscript to be de Quervain tenosynovitis.
Introduction, lines 41-43
de Quervain’s tenosynovitis can be managed nonsurgically through a combination of rest, ice, splinting, and physical therapy. [Source?]
Ans: Thank you for your reminder. We have quoted the reference for this sentence as [3].
Introduction, line 55
Diagnosis is typically made based on physical examination and imaging studies.
COMMENT: What are the gold standard findings and from what examination procedures? (name and findings of orthopedic tests and radiographs)
Ans: Thank you for your reminder. We have added the description as below:” Typical physical examination findings include deep and aching pain, worse pain at night, presentation of pain arc, stiffness and restricted range of motion, and wasting or atrophy of shoulder muscles. Typical imaging study findings include joint space narrowing on shoulder X-rays and capsular thickening change on magnetic resonance imaging or ultrasound studies.”
Introduction, lines 58-60
Redler and Dennis et al. note that steroid injections may also be used to manage symptoms, but caution that these should be used judiciously and only in combination with physical therapy [9].
COMMENT: Why should caution be used? What were the downsides of steroid use noted in the study?
Ans: Thank you for your reminder. We have modified the sentence as below:” Redler and Dennis et al. note that early corticosteroid injection may shorten the duration of symptoms in adhesive capsulitis; however, patients should be aware that anti-inflammatories and corticosteroid injections do not provide a cure but rather alleviate discomfort during necessary physical therapy, while extracorporeal shock wave therapy may be beneficial for diabetic patients experiencing metabolic issues from corticosteroid injection.”
Results, lines 122-123
COMMENT: You wrote "de Quervain’s tenosynovitis" (several times in the Results section) as opposed to other times in the manuscript where you referred to it as de Quervain tenosynovitis. I recommend you stick with one name throughout the manuscript if you insist on writing it out each time (subtle difference, but for the sake of consistency). See tables as well. This should be DQT anyways (see comment above about abbreviation).
Ans: Thank you for your reminder. Because the submission guideline of Healthcare should be at least 4000 words for original article submission. We would like to keep de Quervain tenosynovitis for reaching the minimal word amounts requirement. I have also modified our manuscript to be de Quervain tenosynovitis.
Discussion, line 187
shoulder adhesive capsulitis may...
COMMENT: Shoulder should have a capital S.
Ans: Thank you for your reminder. We have corrected this error.
Discussion, lines 187-191
shoulder adhesive capsulitis may occur on the ipsilateral side of the upper limb in patients with de Quervain’s tenosynovitis due to pain-related limited use of the limb. To compensate for difficulty or pain with hand movement, patients with de Quervain’s tenosynovitis may alter their posture or behavior, further complicating the development of shoulder soreness or stiffness.
COMMENT: Is there any source to support the notion that DQT leads to a lack of use of the shoulder? It is worth noting that you contradict this argument many times indicating that it can lead to overuse of the shoulder.
Ans: Thank you for your reminder. We have modified our sentences as below:” There is no evidence supporting DQT as a cause of or contribution to the cause of shoulder adhesive capsulitis at this point in time. We proposed there possible mechanism relating to this correlation between DQT and shoulder adhesive capsulitis. First, a potential cause of shoulder adhesive capsulitis is the underutilization of the shoulder joint due to limb pain caused by ipsilateral DQT. Patients with DQT may modify their posture or behavior in response to hand movement difficulties or pain, potentially ex-acerbating the progression of shoulder stiffness.”
Discussion, lines 193-194
Both de Quervain's tenosynovitis and shoulder adhesive capsulitis are caused by inflammation of the affected tissues.
COMMENT: This is not true. Inflammation is a symptom of the conditions. Adhesive capsulitis is most likely caused by lack of use of the shoulder (immobility and reduced mobility) and DQT is most likely caused by overuse of the thumb and(or) wrist (repetitive movements, video games).
Ans: Thank you for your suggestions. We have modified this sentence as our second proposed mechanism: “Secondly, while shoulder adhesive capsulitis is commonly attributed to shoulder immobility and reduced mobility resulting from lack of use, and dynamic quadriceps tendonitis (DQT) is often associated with repetitive thumb and/or wrist movements such as those seen in activities like video games, these two conditions might be interconnected through an underlying mechanism involving inflammation that can potentially spread from one area of the body to another.”
Discussion, lines 196-199
Patients with de Quervain's tenosynovitis may compensate for their limited wrist and thumb movement by overusing their shoulder joint, which could lead to inflammation and subsequent shoulder adhesive capsulitis.
and lines 218-220
This rehabilitation may burden the shoulder during movement and increase immobility of the affected upper limb during rest, which may induce shoulder adhesive capsulitis.
and lines 226-230
Patients with de Quervain's tenosynovitis may compensate for their limited wrist and thumb movement by overusing their shoulder joint, which could lead to inflammation and subsequent shoulder adhesive capsulitis. This compensatory movement pattern may persist even after rehabilitation, as patients may continue to use their shoulder joint more than they would otherwise.
and lines 236-239
Patients undergoing rehabilitation for de Quervain's tenosynovitis may be performing exercises and activities that place additional stress on their shoulder joint, which could lead to inflammation and subsequent shoulder adhesive capsulitis.
COMMENT: Adhesive capsulitis is not known for being an overuse injury, but on the contrary, a lack of use injury. This explanation does not make sense as is. Please provide a reputable source.
Ans: Thank you for your suggestions. We have corrected this mistake in our revised discussion section. We think that patients with DQT injured their shoulder from compensation of their upper limb movement during occupation, activity or rehabilitation and limited the use of their shoulder, further disposing development of shoulder adhesive capsulitis.
Discussion, lines 194-195
...it is possible that the inflammatory response may spread from one area of the body to another. [Source?]
and lines 233-235
This increased inflammation and damage to the affected tissues may increase the risk of developing subsequent shoulder adhesive capsulitis. [Source?]
Discussion, lines 252-256
When a distal joint such as the thumb joint is injured and painful, the proximal joint, such as the shoulder joint, is often used to assist with daily activities, which may cause further impingement syndrome or inflammation, further aggravating the development of shoulder adhesive capsulitis. [Source?]
COMMENT: If no source can be found, perhaps you can find any evidence supporting this notion for other comparable areas of the body (ie the big toe and the hip, perhaps in gouty arthritic patients?)
Discussion, lines 281-284
While the exact mechanisms underlying the association between de Quervain's tenosynovitis and subsequent shoulder adhesive capsulitis are not fully understood, there are several preventive rehabilitation procedures that may help reduce the risk of developing shoulder adhesive capsulitis in patients with de Quervain's tenosynovitis.
COMMENT: 1) The exact mechanisms underlying the association between DQT and subsequent shoulder adhesive capsulitis are not understood at all. This appears to be conjecture that is not supported by any evidence and seems contradictory to the evidence of the currently understood mechanism of injury for adhesive capsulitis. 2) There are no preventive rehabilitation procedures that may help reduce the risk of developing shoulder adhesive capsulitis in patients with de Quervain's tenosynovitis. This is the first study investigating this subject. Nothing has been investigated otherwise. Again, it is conjecture. Any recommendations need to be accompanied with a source of previous studies and should be more of a recommendation for future studies involving DQT patients as opposed to recommendations for treatment without evidence to support them.
Ans: Thank you for your suggestions. We have revised our discussion into two main points for these correlations: (1) These two pathologies shared many common risk factors and limited use of shoulder under DQT; (2) Rehabilitation program without considering shoulder conditions. We believe that they are both more convincible than migratory inflammation between the joints. The modified sentences were as below:” Since the causes of correlation between DQT and shoulder adhesive capsulitis are not fully understood, we proposed two possible mechanisms relating to these correlations between DQT and shoulder adhesive capsulitis.
First, a potential cause of shoulder adhesive capsulitis is the underutilization of the shoulder joint due to limb pain caused by ipsilateral DQT. Patients with DQT may modify their posture or behavior in response to hand movement difficulties or pain, potentially exacerbating the progression of shoulder stiffness [22]. The severity of DQT may be positively correlated with the risk of new-onset shoulder adhesive capsulitis because these two diseases share several risk factors. Patients with a history of one condition may be more prone to developing other musculoskeletal conditions, which could increase the risk of shoulder adhesive capsulitis. These may explain that patients with more severe DQT represented by those who required further rehabilitation may have a higher rate predisposing shoulder adhesive capsulitis. When a distal joint such as the thumb joint is injured and painful, the proximal joint, such as the shoulder joint, is often used to assist with daily activities, which may cause further injury of shoulder, such as sprain injury, impingement disease, then further aggravating the development of shoulder adhesive capsulitis [23] .
Secondly, rehabilitation program and conservative treatment of DQT without considering shoulder condition may also play a role in the development of shoulder stiffness . Inadequate rehabilitation protocol was claimed to cause progressive shoulder stiffness after arthroscopic rotator cuff tendon repair [24]. Denard et al. observed that the 6-week immobilization protocol may reduce the incidence of postoperative shoulder stiffness in patients who received shoulder surgery [25]. Rehabilitation of patients with DQT involves protection with a splint or brace and the range of motion protocol of the wrist and thumb. The protection of the upper limb and the rehabilitation program without considering the influence of shoulder movement may induce shoulder adhesive capsulitis based on the above information. Patients who received rehabilitation program without considering shoulder movement may further accidentally sprain the tendons or ligaments of their shoulder joint, causing pain, and further subsequent to limited use of their shoulder joint, causing the development of shoulder adhesive capsulitis and shoulder stiffness. Therefore, specialized shoulder physiotherapy may be recommended for the patients to prevent them from the development of postoperative shoulder stiffness [26]. This could involve incorporating exercises and activities that promote proper posture and mechanics. Additionally, proper supervision and monitoring during rehabilitation may help prevent patients from further injury or inadequately limited use of their shoulder joints [27,28]. We believe that specialized shoulder physiotherapy should be offered to patients with severe DQT who need rehabilitation to prevent the development of additional shoulder problems. Incorporating exercises that specifically target the shoulder joint, including rotator cuff strengthening exercises, shoulder range-of-motion exercises, and shoulder stabilization exercises may help improve shoulder strength and flexibility, reducing the risk of developing shoulder adhesive capsulitis [29]. Occupational therapy interventions, including giving instructions on joint protection and energy conservation, use of assistive devices, and the provision of splints, may be helpful for treatment of DQT and the prevention of further shoulder adhesive capsulitis development [30]. Patients may benefit from education on proper ergonomics and body mechanics during daily activities, particularly those that involve the use of the wrist and hand [31]. This may help reduce the incidental rate of shoulder injury, minimize the risk of inflammation and decrease the risk of developing subsequent shoulder adhesive capsulitis.”
The following exemplifies this comment (Discussion, lines 331-337):
...this study provides important insights into the possible associations between de Quervain's tenosynovitis and subsequent shoulder adhesive capsulitis. The identification of this association provides an opportunity for clinicians to develop preventive programs for shoulder joint health during the rehabilitation of more severe cases of de Quervain's tenosynovitis. Further studies should be conducted to investigate the effects of alterations to rehabilitation programs and occupational therapy on the risk of shoulder adhesive capsulitis among patients with de Quervain's tenosynovitis.
COMMENT: It should also be noted that the presence of DQT provides an opportunity for clinicians to inform patients of the symptoms of adhesive capsulitis and recommend regular shoulder examinations for the purpose of early detection of adhesive capsulitis to maximize implemented treatment outcomes.
Ans: Thank you for your suggestion. We have modified the sentence as below:” The presence of DQT provides an opportunity for clinicians to inform patients of the symptoms of adhesive capsulitis and recommend regular shoulder examinations for the purpose of early detection of adhesive capsulitis to maximize implemented treatment outcomes. The identification of this association between these two pathologies also helps develop preventive programs for shoulder joint health during the rehabilitation for the patients with more severe degree of DQT.”
IMPORTANT: Your proposed mechanism of adhesive capsulitis injury in patients with DQT is extremely important and cannot be unfounded. Sources are needed for your comments throughout the Discussion (see above). And, if there is nothing to support your thoughts, that should be noted. You may need to write something along the lines of:
There is no evidence supporting DQT as a cause of or contribution to the cause of adhesive capsulitis at this point in time. More research is necessary to investigate the relationship between DQT and adhesive capsulitis. To better understand this relationship, future studies need to focus on... This will allow...
Ans: Thank you for your suggestion. We have added below sentence into the last paragraph of our discussion section:” Since there is no evidence supporting DQT as a cause of or contribution to the cause of shoulder adhesive capsulitis at this point in time, future studies also should be conducted to investigate the effects of alterations to rehabilitation programs and occupational therapy on the risk of shoulder adhesive capsulitis among patients with DQT. This will allow us to understand how exactly shoulder and thumb cooperating with each other dynamically in different situations.”
Comments on the Quality of English Language
Amazing job and well-written manuscript. Very few edits regarding the quality of English Language.
Ans: Thank you very much for your encouragement.

Reviewer 2 Report
Thank the authors for submitting the article "de Quesrvain Tenosynovitis As a Risk Factor of New-Onset Adhesive capsulitis: a Nationwide Cohort Study".
I read the work with pleasure and point out some points which characterize submissive work.
Introduction: well done, clear, concise, but focuses on the problem and the study project.
Materials and methods: the number of samples is the aspect that makes the article more attractive to the reader. The numbers and the methodology are remarkable; also in this case, it is clear how we arrived at the two groups.
Results: The complexity of the tables and data is made legible, and I don't see any particular need for correction.
Discussion: this is the part most deserving of a re-reading on your part. In some points, it is repetitive, incredibly long; Between pages 6 and 7, there are some repetitions related to your results and subsequent treatment that could be removed to make the chapter more readable.
Furthermore, in my opinion, one of the main limitations (more important than the others correctly described by you) is not mentioned: the inability to define whether the frozen shoulder is homo or contralateral significantly limits the hypotheses of causality.
You rightly reported some plausible hypotheses that would be strengthened or weakened by the possession of the laterality data of the capsulitis. In my opinion this point should also be emphasized,also, for future investigations.
Conclusion: in general, the findings, as well as the entire object of the study are interesting because they emphasize the possibility of treating one pathology to prevent another, which is always interesting for readers and which marries the journal's mission.
From my point of view, with minimal corrections, an article that can indeed be published in Healthcare journal
Author Response
Thank the authors for submitting the article "de Quervain Tenosynovitis As a Risk Factor of New-Onset Adhesive capsulitis: a Nationwide Cohort Study".
I read the work with pleasure and point out some points which characterize submissive work.
Introduction: well done, clear, concise, but focuses on the problem and the study project.
Materials and methods: the number of samples is the aspect that makes the article more attractive to the reader. The numbers and the methodology are remarkable; also in this case, it is clear how we arrived at the two groups.
Results: The complexity of the tables and data is made legible, and I don't see any particular need for correction.
Ans: Thank you for your positive feedback.
Discussion: this is the part most deserving of a re-reading on your part. In some points, it is repetitive, incredibly long; Between pages 6 and 7, there are some repetitions related to your results and subsequent treatment that could be removed to make the chapter more readable.
Ans: Thank you for your suggestions. We have modified this part. We have revised our discussion into two main points for these correlations: (1) These two pathologies shared many common risk factors and limited use of shoulder under DQT; (2) Rehabilitation program without considering shoulder conditions. We believe that they are both more convincible than migratory inflammation between the joints. The modified sentences were as below:” Since the causes of correlation between DQT and shoulder adhesive capsulitis are not fully understood, we proposed two possible mechanisms relating to these correlations between DQT and shoulder adhesive capsulitis.
First, a potential cause of shoulder adhesive capsulitis is the underutilization of the shoulder joint due to limb pain caused by ipsilateral DQT. Patients with DQT may modify their posture or behavior in response to hand movement difficulties or pain, potentially exacerbating the progression of shoulder stiffness [22]. The severity of DQT may be positively correlated with the risk of new-onset shoulder adhesive capsulitis because these two diseases share several risk factors. Patients with a history of one condition may be more prone to developing other musculoskeletal conditions, which could increase the risk of shoulder adhesive capsulitis. These may explain that patients with more severe DQT represented by those who required further rehabilitation may have a higher rate predisposing shoulder adhesive capsulitis. When a distal joint such as the thumb joint is injured and painful, the proximal joint, such as the shoulder joint, is often used to assist with daily activities, which may cause further injury of shoulder, such as sprain injury, impingement disease, then further aggravating the development of shoulder adhesive capsulitis [23] .
Secondly, rehabilitation program and conservative treatment of DQT without considering shoulder condition may also play a role in the development of shoulder stiffness . Inadequate rehabilitation protocol was claimed to cause progressive shoulder stiffness after arthroscopic rotator cuff tendon repair [24]. Denard et al. observed that the 6-week immobilization protocol may reduce the incidence of postoperative shoulder stiffness in patients who received shoulder surgery [25]. Rehabilitation of patients with DQT involves protection with a splint or brace and the range of motion protocol of the wrist and thumb. The protection of the upper limb and the rehabilitation program without considering the influence of shoulder movement may induce shoulder adhesive capsulitis based on the above information. Patients who received rehabilitation program without considering shoulder movement may further accidentally sprain the tendons or ligaments of their shoulder joint, causing pain, and further subsequent to limited use of their shoulder joint, causing the development of shoulder adhesive capsulitis and shoulder stiffness. Therefore, specialized shoulder physiotherapy may be recommended for the patients to prevent them from the development of postoperative shoulder stiffness [26]. This could involve incorporating exercises and activities that promote proper posture and mechanics. Additionally, proper supervision and monitoring during rehabilitation may help prevent patients from further injury or inadequately limited use of their shoulder joints [27,28]. We believe that specialized shoulder physiotherapy should be offered to patients with severe DQT who need rehabilitation to prevent the development of additional shoulder problems. Incorporating exercises that specifically target the shoulder joint, including rotator cuff strengthening exercises, shoulder range-of-motion exercises, and shoulder stabilization exercises may help improve shoulder strength and flexibility, reducing the risk of developing shoulder adhesive capsulitis [29]. Occupational therapy interventions, including giving instructions on joint protection and energy conservation, use of assistive devices, and the provision of splints, may be helpful for treatment of DQT and the prevention of further shoulder adhesive capsulitis development [30]. Patients may benefit from education on proper ergonomics and body mechanics during daily activities, particularly those that involve the use of the wrist and hand [31]. This may help reduce the incidental rate of shoulder injury, minimize the risk of inflammation and decrease the risk of developing subsequent shoulder adhesive capsulitis.”
We also added two paragraphs regarding to occupational therapy and rehabilitation protocols of de Quervain tenosynovitis to make our discussion more thoroughly and to fit the criteria of Healthcare journal as minimal word counts of submission was 4000 words.
Furthermore, in my opinion, one of the main limitations (more important than the others correctly described by you) is not mentioned: the inability to define whether the frozen shoulder is homo or contralateral significantly limits the hypotheses of causality. You rightly reported some plausible hypotheses that would be strengthened or weakened by the possession of the laterality data of the capsulitis. In my opinion this point should also be emphasized, also, for future investigations.
Ans: Thank you for your suggestions. We have added this into our limitation section as below:” Furthermore, the inability to determine whether the frozen shoulder affects the same side (ipsilateral) or opposite side (contralateral) as the dynamic quadriceps tendonitis (DQT) can greatly impede the establishment of causal hypotheses based on available data. However, obtaining comprehensive information on the progression of both conditions through detailed clinical case studies may better elucidate their interrelationships as functional components of upper limb movement.”
Conclusion: in general, the findings, as well as the entire object of the study are interesting because they emphasize the possibility of treating one pathology to prevent another, which is always interesting for readers, and which marries the journal's mission.
From my point of view, with minimal corrections, an article that can indeed be published in Healthcare journal
Ans: Thank you for your encouragement.

Round 2
Reviewer 1 Report
It seems you did a very thorough job in responding to the previous reviews. Thank you. The manuscript looks great.
1. You spelled out de Quervain tenosynovisits throughout the manuscript but used DQT in the abstract and in the tables and figures. Why didn't you write (DQT) after "De Quervain tenosynovitis" in the Introduction, paragraph 1, line 1 and then, going forward, only use the acronym?
2. Discussion, paragraph 3, line 3
there appears to be a space between stiffness and the period to end the sentence.
Author Response
It seems you did a very thorough job in responding to the previous reviews. Thank you. The manuscript looks great.
- You spelled out de Quervain tenosynovitis throughout the manuscript but used DQT in the abstract and in the tables and figures. Why didn't you write (DQT) after "De Quervain tenosynovitis" in the Introduction, paragraph 1, line 1 and then, going forward, only use the acronym?
Ans: Thank you for your suggestion. We have corrected this part in our manuscript.
- Discussion, paragraph 3, line 3
there appears to be a space between stiffness and the period to end the sentence.
Ans: Thank you for your reminding. We have corrected this error.
